# Eukaryotic Translation Elongation is Modulated by Single Natural Nucleotide Derivatives in the Coding Sequences of mRNAs

**DOI:** 10.3390/genes10020084

**Published:** 2019-01-25

**Authors:** Thomas Philipp Hoernes, David Heimdörfer, Daniel Köstner, Klaus Faserl, Felix Nußbaumer, Raphael Plangger, Christoph Kreutz, Herbert Lindner, Matthias David Erlacher

**Affiliations:** 1Division of Genomics and RNomics, Biocenter, Medical University of Innsbruck, 6020 Innsbruck, Austria; Thomas.Hoernes@i-med.ac.at (T.P.H.); David.Heimdoerfer@i-med.ac.at (D.H.); Daniel.Koestner@i-med.ac.at (D.K.); 2Division of Clinical Biochemistry, Biocenter, Medical University of Innsbruck, 6020 Innsbruck, Austria; Klaus.Faserl@i-med.ac.at (K.F.); Herbert.Lindner@i-med.ac.at (H.L.); 3Institute of Organic Chemistry and Center for Molecular Biosciences (CMBI), University of Innsbruck, 6020 Innsbruck, Austria; felix.nussbaumer@uibk.ac.at (F.N.); raphael.plangger@uibk.ac.at (R.P.); Christoph.Kreutz@uibk.ac.at (C.K.)

**Keywords:** mRNA modification, ribosome, decoding, translation

## Abstract

RNA modifications are crucial factors for efficient protein synthesis. All classes of RNAs that are involved in translation are modified to different extents. Recently, mRNA modifications and their impact on gene regulation became a focus of interest because they can exert a variety of effects on the fate of mRNAs. mRNA modifications within coding sequences can either directly or indirectly interfere with protein synthesis. In order to investigate the roles of various natural occurring modified nucleotides, we site-specifically introduced them into the coding sequence of reporter mRNAs and subsequently translated them in HEK293T cells. The analysis of the respective protein products revealed a strong position-dependent impact of RNA modifications on translation efficiency and accuracy. Whereas a single 5-methylcytosine (m^5^C) or pseudouridine (Ψ) did not reduce product yields, *N*^1^-methyladenosine (m^1^A) generally impeded the translation of the respective modified mRNA. An inhibitory effect of 2′*O*-methlyated nucleotides (Nm) and *N*^6^-methyladenosine (m^6^A) was strongly dependent on their position within the codon. Finally, we could not attribute any miscoding potential to the set of mRNA modifications tested in HEK293T cells.

## 1. Introduction

The idea that mRNAs are simple, static carriers of a plain four-letter code was challenged by the discovery of an increasing number of modified nucleotides within mRNAs (reviewed in [1,2]). These modifications, found in the untranslated regions (UTRs) as well as in the coding sequences (CDSs), have the capability to add an additional layer of information to the genetic code, thereby expanding the organisms’ toolkit of regulating gene expression. Due to novel refinements in mass spectrometry (MS) and high throughput sequencing, their number and their respective positions seem to change constantly. So far, *N*^6^-methyladenosine (m^6^A) [3,4,5,6], 5-methylcytosine (m^5^C) [7,8,9], 5-hydroxycytosine (hm^5^C) [10,11], pseudouridine (Ψ) [12,13,14], *N*^1^-methyladenosine (m^1^A) [15,16,17], *N*^3^-methylcytosine (m^3^C) [18], and 2′*O*-methlyated nucleotides (Nm) [19,20] have been reported to decorate eukaryotic mRNAs. However, recently, the prevalence of some modifications within mRNAs was challenged by the reevaluation of some of the already published datasets [6,17,21,22,23]. In addition to the current discussion about the number and the positions of the identified modifications, their significance and the role of numerous modifications are also currently debated [24,25].

So far, modifications have been described to influence mRNA processing [26,27], localization [28,29], stability [30,31], and also local secondary structures [32,33,34]. Protein synthesis, i.e., the last step of gene expression, can also be affected by various nucleotide derivatives. Inhibitory as well as stimulatory effects through mRNA modifications have been reported depending on their type and position (reviewed in [35]). Whereas mRNAs harboring randomly distributed m^5^Cs and Ψs provided higher yields of the respective proteins, multiple m^6^As were strongly repressive [36]. In bacteria, single m^6^As within the CDS reduced protein yields, dependent on the sequence context and the position of the modification within the codon [37,38]. Yet, a single m^6^A is also able to stimulate translation, when located within the 5′ UTR [39,40]. The proposed roles of m^1^A are similarly diverse. Thousands of methylation sites have been proposed to be located around the start codon. In addition, ribosome profiling revealed m^1^A to be generally mildly stimulatory [15]. On the contrary, a recent study not only questioned the number of m^1^As within mRNAs, but also described an inhibitory effect of m^1^A on protein synthesis [17]. Also, Nms within the CDS were reported to strongly interfere with bacterial translation [37]. However, in yeast, hundreds of sites have been identified, and ribosome profiling did not reveal any detectable ribosomal stalling events at the reported Nm sites [20]. In addition, in HEK293 and HeLa cells, thousands of methylation sites have been postulated, many of them located within mRNAs [19].

Since the investigations on the impact of mRNA modification have been carried out in different translation systems, ranging from bacteria to mammals, and from in vitro to in vivo systems [15,16,17,36,37,38,39,41,42], we strived to systematically investigate the impact of mRNA modifications in HEK293T cells on a distinct step during protein synthesis, namely translation elongation. We aimed to further clarify the potential roles of various modifications in human cells and to bring our results in context with already described observations. To do so, we introduced single modifications specifically into one codon of a reporter mRNA and determined the quantity and the quality of the translation products by Western blotting and MS analyses, respectively. In analogy to bacterial translation systems, the effects on translation efficiency and accuracy were strongly dependent on the position of the modification within the codon. Our results indicate that mRNA modification can serve, in principal, as an efficient measure to impact gene expression post-transcriptionally. However, whether and to what extent this regulatory strategy is indeed exerted in various organisms remains elusive.

## 2. Materials and Methods

### 2.1. Sequences

The template for the T7 RNA transcription was generated through PCR amplification on the eGFP cassette of the lentiviral pHR-DEST-SFFV-eGFP plasmid, introducing the N-terminal Flag-tag employing the forward primer 5′-GCTCTAGA*TAATACGACTCACTATA*GGGGGCCACC**ATG**GACTACAAGGACGACGACGATAAGGTGAGCAAGGGCGAGG-3’ (T7 promoter italicized, start codon in bold, and FLAG-tag underlined) and the reverse primer 5′-mCmGTCCTCCTTGAAGTCGATGCCCTTCAGCTC-3′. The transcript was then ligated to the respective poly(A)-tailed oligonucleotides yielding the Cap-FLAG-eGFP-ErmCL-poly(A) [43].

### 2.2. Oligonucleotide Synthesis

Oligonucleotides harboring m^6^A, m^5^C, Ψ, and 2′*O* methyl groups were obtained from Integrated DNA Technologies (IDT, Coralville, IA, USA) and Dharmacon (Lafayette, CO, USA) [37,44]. Oligonucleotides harboring m^1^A were synthesized, deprotected, and tested for quality, as previously described [45,46].

### 2.3. Transcription of Capped and Fully Modified mRNAs

The template for transcription was generated through PCR employing the reverse primer (5′-TTACTTGTACAGCTCGTCC-3′). Fully m^5^C- or Ψ-modified and capped transcripts were generated by employing the HiScribe T7 High Yield RNA Synthesis kit (New England Biolabs, Ipswich, MA, USA), as described by the manufacturer. Cytidine or uridine were quantitatively replaced by m^5^C or Ψ (f.c. 1.25 mM; TriLink, San Diego, CA, USA). The Anti Reverse Cap Analog (ARCA) was purchased from TriLink, and was co-transcriptionally incorporated (f.c. 4 mM).

### 2.4. Splinted mRNA Ligation

Cap-Flag-eGFP-ErmCL-poly(A) reporter mRNAs were generated by ligating the capped 5′-transcript to the poly(A)-tailed ErmCL oligonucleotide 5′-P-AUUAUNNNCCAAACAAAAAA**UAA**-3′ (The sense codon that was modified or exchanged is underlined; stop codon in bold) bridged by splinter 5′-TTTTTTGTTTGGNNNATAATCGTCCTCCTTGAAGTCGATG-3′ (the underlined sequence was adjusted to be reverse complementary to the investigated codon), employing T4 RNA ligase 2 (NEB) as described previously [43,44,47]. Ligation products were purified by employing a magnetic mRNA isolation kit (NEB). mRNA purity and integrity were checked with a 2100 Bioanalyzer (Agilent, Santa Clara, CA, USA). To address the effects of a Nm, a poly(A)-tailed oligonucleotide encoding the 5-HT_2C_R mRNA sequence 5′-P-UAGCAAUACGUAAUCCUAUUGAGCAUAGC**UAA**-3′ (The methylated site is underlined; UAA stop codon in bold) was ligated to the capped FLAG-eGFP mRNA fragment.

### 2.5. Cell Culture, Transfection and Western Blotting

This procedure was basically carried out as previously described [43]. 40% confluent HEK293T or N2a cells were transfected with 10 pmol of the respective mRNAs using metafectene (Biontex, München, Germany). Twenty-four hours after transfection, cells were lysed and the protein isolation quantified via the Bradford assay. Exactly 20 µg of total protein were separated by SDS-PAGE and blotted to 0.45 µm PVDF membranes (GE Healthcare, Chicago, IL, USA). The blots were probed with an anti-Flag M2 antibody (Sigma, Saint Louis, MO, USA, 1:3000 dilution) or an anti-α tubulin antibody (Abcam, Cambridge, UK, 1:7000) overnight at 4 °C. As a secondary antibody, a goat anti-mouse HRP-conjugated antibody (Dako, Glostrup, Denmark) was employed in a 1:3000 dilution. The blot was developed using the Pierce ECL Western blotting substrate (Thermo Scientific, Waltham, MA, USA).

### 2.6. Mass Spectrometry Analysis of Translation Products

Flag-eGFP peptides translated in HEK293T cells were purified with anti-Flag M2 magnetic beads (Sigma) [43,44]. Pulled down proteins were washed with 50 mM ammonium acetate, and directly digested on the beads in an ammonium bicarbonate buffer (100 mM, pH 8.0). Proteins were reduced with dithiothreitol (10 mM) for 30 min at 56 °C, digested for 6 hr at 37 °C by adding 0.5 µg trypsin, and alkylated with iodoacetamide (55 mM) at room temperature for 20 min.

Peptides were analyzed using a Dionex, UltiMate 3000 nano-HPLC system (Germering, Germany) coupled via nanospray ionization source to a Thermo Scientific Q Exactive HF mass spectrometer (Vienna, Austria) using instrument settings as described previously [48]. In brief, peptides were separated on a homemade fritless fused-silica capillary column (100 µm i.d. × 20 cm length) packed with 2.4 µm reversed-phase material (ReproSil-Pur C18-AQ with 120 Å pores). The gradient (solvent A: 0.1% formic acid; solvent B: 0.1% formic acid in 85% acetonitrile) started at 4% B, for 4 min. The concentration of solvent B was then increased linearly from 4% to 35% over 53 min, and from 35% to 100% over 5 min. A flowrate of 250 nL/min was applied. Mass spectra were acquired in positive ion mode applying data dependent acquisition mode. Survey MS spectra (*m*/*z* 300–1750) were acquired with a resolution of R = 60,000 at an AGC target of 1 × 10^6^. To generate MS/MS spectra, the 20 highest precursors were selected for higher-energy collisional dissociation (HCD), applying a normalized collision energy of 28.0. Fragments were scanned with a resolution of R = 30,000 at an AGC target of 5 × 10^5^. All scans were acquired in profile mode at a maximum ionization time set to 120 ms.

Database search was performed using ProteomeDiscoverer (Version 2.1, Thermo Scientific) with search engine Sequest HT. MS/MS spectra were searched against a human protein database (Uniprot, reference proteome, last modified Feb 2018, 20,939 entries) to which 21 different ErmCL protein sequences were added. All mass spectrometry data have been deposited to the ProteomeXchange Consortium via the PRIDE database with the data set identifier PXD011860 [49].

## 3. Results

The present study aimed to investigate the potential of some described mRNA modifications (Figure 1A) to intervene with translation elongation. By applying an RNA ligation strategy capped and polyadenylated mRNAs, carrying a defined number of modifications at distinct positions, were generated. In order to reduce the complexity, and to be able to compare the effects, the same codon at position 145 of the reporter mRNA was modified. Whereas so far, the modified mRNAs were predominantly tested in various eukaryotic in vitro translation systems, in this study HEK293T cells were employed. The mRNAs were transfected and after 24 h, total protein extracts were prepared. The respective translation products were analyzed for their quantity and quality by Western blotting and MS, respectively.

### 3.1. Effects of mRNA Modifications on Eukaryotic Translation Efficiency

In 2014, several hundreds of yeast and human mRNAs were identified to harbor Ψs [12,14]. The presence of multiple Ψs within eukaryotic mRNAs was reported to increase the yield of the respective protein products [36]. Consequently, the presence of a single Ψ should not impede translation, but it might already be sufficient to provide a stimulatory effect, independent of its location. We introduced Ψ separately at the first, second, and third nucleotides of the phenylalanine (Phe) codon UUU and determined the resulting translation efficiency by Western blotting. Indeed, a single Ψ was not sufficient to either stimulate or impede translation elongation (Figure 1B).

m^5^C within an mRNA was also described to increase the protein yields, especially if it was present at multiple positions [36]. Located close to the initiation site, already a single m^5^C could stimulate gene expression by the binding of ALYREF, leading to an increased export of the modified mRNA from the nucleus to the cytoplasm [29]. In order to test if one m^5^C within a CDS can directly affect translation elongation, a m^5^C was separately introduced in codon 145. Therefore, the UUU codon was changed to a CCC, encoding for proline, and m^5^C was introduced at all three codon positions separately. Interestingly, a single m^5^C within the CDS did not stimulate the translation of the modified mRNA (Figure 1C).

As Ψ and m^5^C do not alter the Watson–Crick edge during formation of the codon–anticodon interaction, it seems conceivable that they do not strongly impede translation elongation. In contrast, m^1^A harbors a methyl group directly at the Watson–Crick edge, possibly preventing the formation of the codon-anticodon helix (Figure 1A). Indeed, m^1^A within a lysine codon (AAA), did not allow protein synthesis at all (Figure 1D). This distinct inhibition was observed independent of the location of m^1^A within the codon. Even at the wobble position, which is usually less restrictive in terms of base pairing geometry [50,51], the methylation hindered protein synthesis.

m^6^A within mRNAs was reported to strongly inhibit translation in bacteria but also in eukaryotic translation systems, especially when m^6^A was present at multiple positions [36]. In our study, already a single m^6^A impeded translation in HEK293T cells, especially if located at the first codon position (Figure 1E). At the second and third position the inhibitory effect was less distinct (Figure 1E).

In addition to the incorporation of single modifications, mRNAs carrying multiple modifications were tested. Since Ψ and m^5^C did not alter the yields of protein synthesis, we co-transcriptionally incorporated them in vitro resulting in full substitutions of U and C by Ψ and m^5^C, respectively (Figure 2A). This resulted in more than 50 modified bases in many sequence and codon combinations. The fully substituted Ψ and the fully substituted m^5^C mRNAs affected the efficiency of translation only modestly (Figure 2B). In case of m^6^A we limited the number of m^6^A within the CDS since already a single methylation strongly interfered with protein synthesis. The presence of two m^6^As in close distance or three consecutive m^6^As within one codon, completely abolished translation elongation (Figure 2C).

Nm showed a strong inhibitory effect on bacterial translation, but this type of methylation was recently reported to be present at multiple positions within eukaryotic mRNAs [19,20,37,38]. We generated mRNAs carrying single methylation sites within the CDS and observed an inhibition of translation at all three codon nucleotides, showing the strongest reduction of protein yield at the second codon nucleotide (Figure 3A). To exclude a sequence or codon specific effect, we substituted the modified sequence with a 10-codon fragment of the human serotonin 5-HT_2C_ receptor (5-HT_2C_R) mRNA, which has been proposed to be methylated by a small nucleolar RNA (snoRNA)-guided protein complex at the second codon position [52]. Also, within this sequence context the Nm fully sequestered translation elongation of the modified mRNA in HEK293T cells (Figure 3B). The methylation of the 5-HT_2C_R has been proposed to be guided by the brain-specific C/D box snoRNA HBII-52 (SNORD115) [53]. Therefore, we also analyzed a mouse neuronal cell line, i.e., N2a cells, for the ability to translate the methylated 5-HT_2C_R sequence. In line with the results of the HEK293T cells the respective translation product could not be observed (Figure 3B).

### 3.2. Effects of mRNA Modifications on Eukaryotic Decoding

Since mRNA modifications within the CDS of mRNAs can modify translation efficiency, it is also conceivable that translation accuracy is affected. Therefore, the respective proteins were purified employing anti-Flag magnetic beads and subjected to MS analysis. As Ψ was already reported to cause stop codon read-through by recoding stop codons into sense codons [56,57], Ψ was a promising candidate for rewiring the genetic code. However, at least within an UUU codon single Ψs did not induce any detectable recoding events (detection limit ~1%). In case such a recoding event might depend on the codon or the sequence context, we expanded our analysis. We substituted all Us within the reporter mRNA by Ψs and analyzed multiple amino acids of the respective protein product encoded by Ψ containing codons (Figure 2A). Also, this extended analysis of 13 additional codons harboring Ψs at the first and second nucleotide of the codon did not reveal any miscoding event.

In addition, m^5^C was tested for its ability to interfere with decoding, since in bacterial systems a single m^5^C at the second nucleotide of the CCC codon caused an enhanced incorporation of leucine instead of proline [37]. Because of the potentially higher translation accuracy of the eukaryotic translation machinery [58,59,60], miscoding events seemed unlikely. MS analysis of the peptide products resulting from mRNAs carrying single as well as multiple m^5^Cs (9 codons) did not reveal any increased tendency for misincorporations (Figure 2A). Also, in case of m^6^A or Nm no recoding events were detectable, which is in line with earlier studies in bacterial but also in eukaryotic in vitro systems [37,42].

## 4. Discussion

mRNA modifications have numerous effects on gene expression. These effects are dependent on the type, location, and the number of the modifications within the mRNA. They can impact almost every step during the lifetime of an mRNA, as they interfere with mRNA splicing, export, stability, and also its translation [61]. Concerning protein synthesis, numerous different effects were reported in various translation systems [35]. In order to reduce the complexity, we were mainly interested in the impact of several reported mRNA modifications on translation efficiency and accuracy in HEK293T cells. Thereby, we focused our studies on a set of selected codons. Thus, we cannot exclude the possibility that specific modification within other codons could differently affect translation, or to some extent, the mRNA stability.

Ψ was among the first identified RNA modifications and was found to be present at hundreds of sites within mRNAs [12,13,14]. Multiple Ψs were already reported to be poorly translated in vitro, by using wheat germ extracts and also bacterial translation systems [36,37]. Strikingly, translation of pseudouridylated mRNAs in rabbit reticulocytes extracts and in different mammalian cell lines led to increased protein yields [36,62]. A single Ψ within the CDS of our reporter mRNA was not sufficient to reduce or stimulate the translation efficiency (Figure 1B). In terms of translation accuracy, Ψs did not cause any detectable miscoding events. Neither a single Ψ within the UUU codon nor multiple Ψs in other tested codons caused a rewiring of the genetic code (Figure 2A). This was to some extent unexpected, since Ψ was considered a promising candidate to expand the genetic code by partially rewiring it [56,63]. Since we could not investigate all possible sequence combinations of Ψ in all CDSs, it is still possible that in some codon or sequence contexts Ψ can lead to miscoding. However, we propose that these rewiring events would be limited to specific sequence contexts rather than representing a general attribute of Ψ within mRNAs.

m^5^C, in analogy to Ψ, was reported to be able to enhance eukaryotic translation [36]. This effect could not be observed by introducing a single m^5^C within a proline codon into the coding sequence (Figure 1C). In addition, these modified proline codons did not induce amino acid misincorporations, which contradict the findings from bacterial systems [37]. An extended analysis of additional codons at different positions within our reporter mRNA did not reveal miscoding events at any of the tested codons (Figure 2A).

Strikingly, contradicting earlier observations, our eGFP reporter mRNA carrying multiple m^5^Cs or Ψs did not result in higher translation efficiency (Figure 2B). It was shown that such modified mRNAs would reduce immune activation, and therefore cause higher product yields. However, this effect strongly depends on the employed mRNA sequences, the UTRs, and the purification procedure after in vitro transcription [62,64,65]. The translation process itself does not seem to be discernibly affected by the presence of numerous m^5^Cs or Ψs within the CDS, since neither one nor multiple modifications reduced the product yields (Figure 1B,C and Figure 2B).

m^1^A has also been identified to decorate mRNAs. The first study describing its presence within mRNAs reported more than 7000 sites, mainly located around the start codon and reported an enhanced translation of the m^1^A-modified mRNAs [15]. Recently, the number of methylation sites was questioned by a refined analysis, and only 12 sites, of which only three were detected without preceding enrichment, were proposed [17]. The same study also observed a reduced translation of m^1^A-modified mRNAs. Indeed, already, a single m^1^A within the CDS inhibited translation independent of its position within the codon (Figure 1D). This can be rationalized by a steric clash by the methyl group during the formation of the codon–anticodon interaction. In addition, the presence of a H-bond between the *N*^1^ (pyrimidine) and *N*^3^ (purine) is also essential during decoding, and it cannot be formed by m^1^A [44]. Although we cannot exclude the stimulatory effect of m^1^A when located within the 5′ UTR, it seems likely that within the CDS m^1^A is a strong inhibitor of translation elongation.

m^6^A and Nm do not (directly) interfere with the formation of the Watson–Crick geometry, but they also substantially impede translation elongation (Figure 1E and Figure 3A,B). These effects were, in contrast to m^1^A, strongly dependent on the position of the modification within a codon. m^6^A showed the strongest inhibitory effect when located at the first codon position (Figure 1E). In bacteria, m^6^A in the first codon position can induce minor local perturbations within the decoding site leading to a destabilization of the codon-anticodon helix, which might also be true for eukaryotic decoding [41]. As m^6^As are rather abundant in eukaryotic mRNAs [5,66], they might also be attributed with a regulatory role during translation. It has been shown that m^6^A within the 5′ UTR stimulates translation initiation of uncapped mRNAs [39]. Thereby, m^6^A can promote initiation independent of certain initiation factors [38]. These findings underline the potentially important role of this modification during gene expression. However, with respect to translation elongation, m^6^A appears to be inhibitory, and strongly dependent on the position within the codon as observed in prokaryotic [37,41] and eukaryotic translation systems [36]. The impeding effects of m^6^A were also additive (Figure 2C), providing an explanation for why mRNAs harboring multiple m^6^As (>5%) could not be translated in bacteria and eukaryotes [36].

Whereas m^6^A showed the strongest inhibition at the first nucleotide of the codon (Figure 1E), Nm completely impeded elongation when located at the second nucleotide (Figure 3A,B). Recently, hundreds of Nms were revealed in yeast, many of which are located in CDSs [20]. Similarly, a screen in HeLa cells postulated thousands of sites predominately found in CDSs [19,21]. It could be argued that stalling events are specific for the methylated codons that were screened in this study and the conclusions cannot be generalized. However, we believe that this is highly unlikely, since this strong inhibition of elongation could also be observed in bacteria [37]. In bacteria, the inhibitory effect of Nm is most likely the result of a steric clash of the methyl group with the nucleotide A1492 of the 16S rRNA [37,38]. Since the decoding site is highly conserved, it is feasible that the Nm also interferes with the eukaryotic counterpart of A1492, namely A1824 (Figure 3C).

A general inhibitory function of Nm raises the possibility of C/D box snoRNAs being potentially involved in regulating gene expression through interfering with translation. Since the targets of many snoRNAs are still unknown, it seems feasible that some of them might function as regulators of protein synthesis by guiding the ribonucleoprotein complex to mRNA target sites, leading to methylations within CDSs.

During the preparation of the manuscript, *N*^4^-acetylcytidine was revealed to be present within eukaryotic mRNAs. This novel modification was described to stimulate translation, especially when it is located at the wobble position of a codon. This discovery adds another piece to the rather complex regulation of gene expression through mRNA modifications [67].

Our data indicate that in principal, single modifications can serve as potent regulators of gene expression. Their effects are not only dependent on the type of the modification, but also on their position within the codon. Strikingly, despite earlier observations indicating a putative recoding potential of mRNA modifications, the tested nucleotide derivatives did not interfere with eukaryotic decoding. Whether endogenous mRNA modifications at low stoichiometries exert a direct regulation of translation in vivo remains elusive and will be subject of future studies.

## Figures and Tables

**Figure 1 genes-10-00084-f001:**
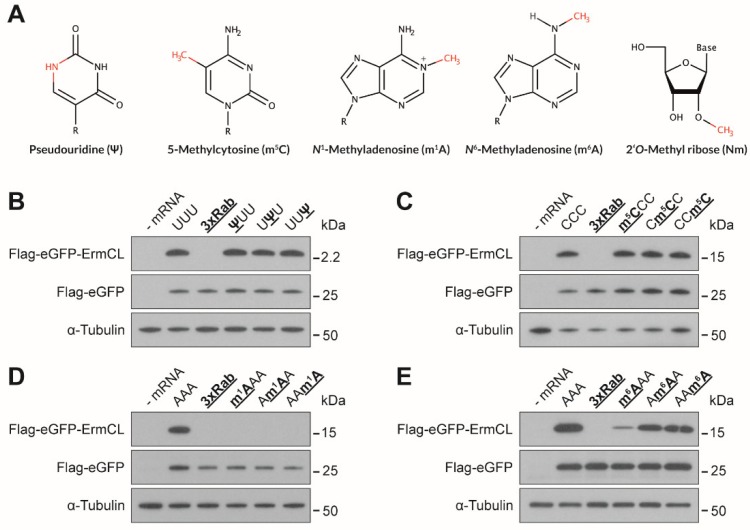
Translation efficiency and accuracy of site-specifically modified mRNA nucleotides. (**A**) The nucleotide derivatives that were investigated are depicted. The chemical groups differing from the standard nucleotides are displayed in red. (**B**–**E**) Western blot analyses of modified mRNAs harboring (**B**) pseudouridine (Ψ), (**C**) 5-methylcytosine (m^5^C), (**D**) *N*^1^-methyladenosine (m^1^A), and (**E**) *N*^6^-methyladenosine (m^6^A) translated in HEK293T cells. An unmodified eGFP mRNA was employed as an internal transfection control and α-tubulin as a loading control.

**Figure 2 genes-10-00084-f002:**
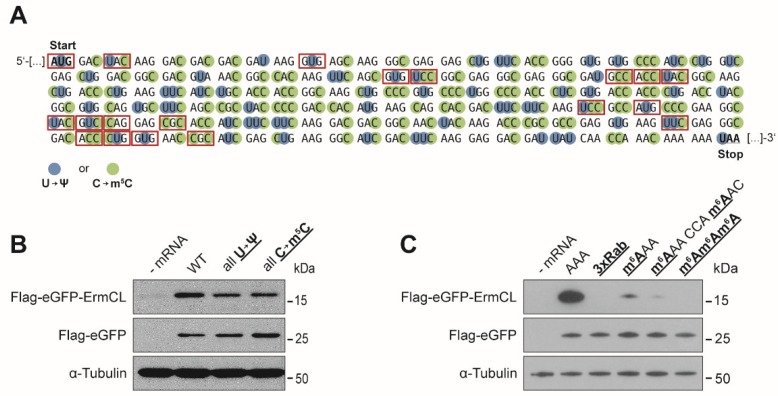
Translation efficiency and accuracy of multiple modified mRNA nucleotides. (**A**) Sequence of the reporter eGFP mRNA is depicted. Employing in vitro transcription, the Us and Cs were substituted with Ψs and m^5^Cs, respectively. The respective Cs are depicted in green and the Us in blue. The amino acids corresponding to selected codons (red boxes) were analyzed by mass spectrometry for their identity (detection limit ~1%). (**B**) Western blot analysis of the translation products derived from mRNAs carrying multiple Ψs or m^5^Cs. (**C**) Western blot analysis of translation products harboring one, two or three m^6^As.

**Figure 3 genes-10-00084-f003:**
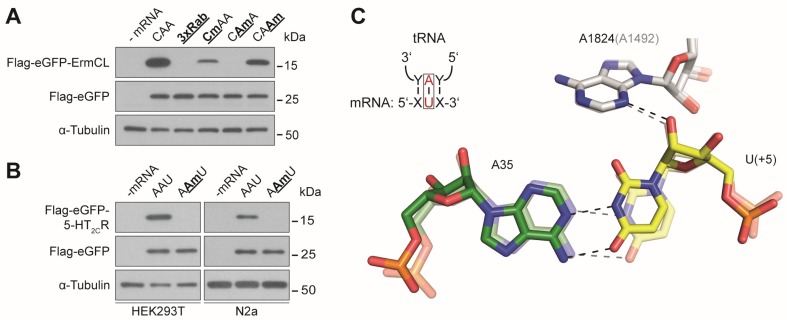
Effect of Nm on translation elongation. (**A**) Western blot analysis of translation products carrying Nm at the codon 147 of the reporter mRNA and (**B**) at the second nucleotide of the AAU codon within the sequence context of the 5-HT_2c_R mRNA. (**C**) Structure of the second base pair of the codon-anticodon interaction and the interaction with A1824 (A1492) of the 18S rRNA (16S rRNA) of the 40S (30S) ribosomal subunit. The eukaryotic decoding center is depicted in the foreground (modified from [54]), the *E. coli* decoding center in the background (modified from [55]).

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
