# Peer review of "Eukaryotic Translation Elongation is Modulated by Single Natural Nucleotide Derivatives in the Coding Sequences of mRNAs"

_genes, 2019, doi:10.3390/genes10020084_

Round 1

Reviewer 1 Report

The article titled “Eukaryotic translation is modulated by natural mRNA nucleotide derivatives” by Hoernes, Heimdorfer et al. have investigated the effect of naturally  occurring chemical modifications of mRNA nucleotides such as m6A, m1A,  m5C, pseudouridine and 2’-Ome in mammalian translation in vivo. The authors synthesized mRNA that either  carries chemically modified nucleotide at specific position or at all  U’s or C’s in the case of pseudouridine or m5C, transfected the  mammalian cell (HEK293T) with modified mRNA, and assayed  how the presence of the modifications affect translation elongation.  Using different methods of mass spectrometry and western blotting  assays, both qualitative and quantitative effects of mRNA modifications  were tested via well-designed experiments. While  the majority of effects are comparable to effects observed in bacterial  system (Hoernes et al. NAR 2015), observation of lack of miscoding  induced by pseudouridine as well as m5C are surprising.

In general, this is a valuable work that supports authors’ previous finding in the bacterial system holds for the mammalian  system in vivo, possibly due to the conserved mechanism  of translation elongation. Authors’ claims throughout the manuscript  are well-supported by the results.

Major comments:

1.     While  the effects observed for each modification are convincing, could the  authors comment on the stability of transfected mRNAs and how  modifications such as pseudouridine  or m5C might have effect on translation efficiency via stabilizing the  mRNA against degradation machinery?

2.     The  representation of Figure 1F is a bit confusing, as all of modifications  have no miscoding effect (unlike similar plots used in the authors’  previous papers). Zooming  into the 95-100% region to show error-bars or simply stating that none  shows the miscoding in a given detection limit in the figure legend may  be helpful.

3.     Was  there any stop-codon read-through on the UAA codon, detected in the  mass-spec analysis? According to ref. 54 and 55, it may have been read  by Thr- or Ser-tRNA, and allow  direct comparison with the previous results.

4.     The  use of “quantitative” and “qualitative” for the result section 3.1 and  3.2 needs to be revised. Currently, quantitative MS analysis is used to  detect “qualitative” effects  (more accurately modification-induced miscoding events) in protein  synthesis, and qualitative western-blot assay is used to detect  “quantitative” effects in protein synthesis (but to argue this,  quantifications of intensities within the western-blot are necessary).

Minor comments:

Figure 2A: What does the red box stands for?

p.3  Splinted mRNA ligation: In case NNN were not AAA (CCC or UUU),  different splinter has been used or other correct base-pairing  were enough to drive ligation?

p.5 3rd paragraph: reference 36 (about pseudouridine) seems out of place for m6A. Perhaps reference 37?

p.5 4th paragraph: Briefly comment on the incorporation of pseudouridine/m5C in in vitro transcription system, and how full products have been selected before ligation.

Author Response

Please find our detailed point to point response attached.

Reviewer 2 Report

In this manuscript, Hoernes and colleagues explore the relationship between mRNA modifications and translation. Focusing on selected modifications that have been detected in mRNA and previously implicated in translational regulation, the authors generated a series of reporters primarily for transfection into HEK cells. Two general approaches were utilized: 1) site specific incorporation of modified residues through splint ligation and 2) randomized incorporation through in vitro transcription. The first approach, in particular, represents a major advance for allowing direct comparison of distinct modifications at analogous locations within the reporter. Relatedly, this approach allowed for specific modification at codon position 1, 2 or 3 and examination of the net impact on translation. While this strategy facilitates the direct comparison of distinct mRNA modifications in a homologous setting, the authors acknowledge that there are significant limitations that hamper broad interpretations. Most notably, select codons may be more or less sensitive to modification or may only be sensitive within specific contexts. Nonetheless, the authors uncovered intriguing insights relating modification status and codon position to perturbed translation. The directness of these comparisons constitute a significant advance.

I only have a couple comments and suggestions:

Comments:

1)   It is not clear whether the various RNAs are equally stable upon transfection into HEK. Variations in stability could affect downstream results on translation. For the purposes of this manuscript, a simple qRT-PCR comparison of mRNA recovery post-transfection would help. If impossible at this stage, the potential for mRNA degradation should be clearly specified. 

2)   The mass spec results regarding fidelity are difficult to decipher. The text suggests no variation, but a single data panel is shown. Some additional explanation is required to solidify this important point.

Suggestions: 

1)   To get a sense of overall expression, how much protein lysate was used for the western blot?

2)   Some mention of how the tRNA pool within HEK could influence these results should be made.

Overall, this manuscript represents an important contribution to the field. 

Author Response

(The authors gave the same response as above.)

Reviewer 3 Report

RNA contains a large number of posttranscriptional modifications. In the manuscript titled with’ Eukaryotic translation is modulated by natural mRNA nucleotide derivatives’, the authors investigate the impact of mRNA modifications  in HEK293T cells on translation. The authors showed that mRNA  modification  can  serve  as regulators of gene expression and translation products could be analyzed for their quantity and quality by Western blotting and MS. Overall the manuscript fits the scope of the journal. The references also reflect the recent progress.

The following are my major comments: 

1 Regarding the translation efficiency and accuracy  RNAs, the authors should add more details about the MS analysis in the method section.

2 N6--methyladenosine (m6A) is one of the most abundant internal mRNA modification. When it is located within the 5’UTR m6A is able to stimulate translation. What is the role of m6A on translation initiation?  The authors should discuss the possible underlying regulatory mechanisms and their biological consequences base on their results.

3 The authors should consider making the tile more specific.

4. Fig 1(F) is confusing. The authors should make it more readable.

Author Response

(The authors gave the same response as above.)

Round 2

Reviewer 3 Report

The revision of the manuscript is of very good quality and the title of the manuscript is much more specific. The authors have successfully addressed my comments and my review is generally positive. I wish to congratulate the authors for the excellent work and I would suggest publishing this revised version in this special issue on RNA modification in Genes.